

# Challenges to Implementing Bottom-Up Flood Risk Decision Analysis Frameworks: How Strong are Social Networks of Flooding Professionals?

James O. Knighton[1], Osamu Tsuda[2], Rebecca Elliott[3], M. Todd Walter[1]

[1]Department of Biological and Environmental Engineering, Cornell University, Ithaca, 14850, US
[2]Department of Architecture, Art, and Planning, Cornell University, Ithaca, 14850, US
[3] Department of Sociology, London School of Economics, London, WC2A 2AE, UK

*Correspondence to*: James O. Knighton (jok8@cornell.edu)

**Abstract.** Recent developments in bottom-up vulnerability-based decision analysis frameworks present promising opportunities for flood practitioners to simplify complex decisions regarding risk mitigation and climate adaptation. This family of methodologies relies on strong social networks among flood practitioners and the public to support careful definition of stakeholder relevant thresholds and vulnerabilities to hazards. In parallel, flood researchers are directly considering distinct

atmospheric mechanisms that induce flooding to readily incorporate information on future climate projections. We perform a case study of flood professionals actively engaged in flood risk mitigation within Tompkins County, NY US, a community dealing with moderate flooding, to gage how much variance exists among professionals from the perspective of establishing a bottom-up flood mitigation study from an atmospheric perspective. Results of this case study indicate disagreement among flooding professionals as to which socio-economic losses constitute a flood, disagreement on anticipated community needs,

weak understanding of climate-weather-flood linkages, and some disagreement on community perceptions on climate adaptation. In aggregate, the knowledge base of the Tompkins County flood practitioners provides a well-defined picture of community vulnerability and perceptions. Our research supports the growing evidence that collaborative interdisciplinary flood mitigation work could reduce risk, and potentially better support the implementation of emerging bottom-up decision analysis frameworks for flood mitigation and climate adaptation.

**1 Introduction**

Societal vulnerability to flooding is a complex function of physical hydrological processes, overlaid with our economic relationship to the land (Wheater & Evans, 2009), community perceptions and responses to risk (e.g. Vinh Hung et al. 2007), and the fundamental ability of experts to clearly communicate these risks to facilitate decisions by policy makers (Pappenberger et al., 2013). Recent flood losses across North America, Europe, and Asia have been exacerbated by an imperfect understanding

of hazards (Merz et al., 2015), fundamental issues in how governmental organizations store and leverage data (Lane et al.,





2011; Harries et al., 2011), the cognitive biases of individuals (Merz et al., 2015; Harries et al., 2008), socially organized apathy (Norgaard, 2011), and governmental response to societal deviations from anticipated rational behavior (Lupton, 2013). Flood risk analysis is inherently difficult due to the infrequency of flooding events (Merz et al. 2015), a globally non-stationary climate leading to more extreme precipitation (Trenberth et al. 2011), often non-linear hydrologic rainfall-runoff responses

(e.g. Mathias et al. 2016), and complex human-flood interactions (e.g. Collenteur et al. 2015) all of which can act to limit the intuition of decision makers for understanding flooding risks and selecting mitigation options (Merz et al. 2015). For any given local flood risk, different data, models, and assumptions can be combined in various ways to yield alternative, reasonable measures of the "true" or "real" risk.

Flooding governance is typically discussed as being "top-down" or "bottom-up." Top-down typically refers to a technocratic

hierarchy, often in the form of national institutions acting as the sole decision makers (e.g. Plate, 2002). These approaches have been associated with a reliance on hazards-based assessments of risk. Bottom-up approaches, in contrast, leverage the knowledge, experiences, preferences, and vulnerabilities of end-users explicitly in problem definition and selection of mitigation actions. In bottom-up approaches, decision making is generally a collaborative process across institutions. Previous work has shown that flood risk mitigation can benefit from a combination of top-down and bottom-up approaches where

decisions are collaboratively refined and implemented across institutions (Serra-Llobet et al. 2016), often with direct input from stakeholders (e.g. Edelenbos et al. 2017; Knighton et al. 2017a). Pahl-Wostl et al. (2013) demonstrate through a case study of three European nations, the relative benefits of different governance schemes. Top down mitigation, as found in Germany, allowed for clearer roles in decision making, centralized repositories of knowledge, and more rapid action within limited windows of opportunity. The vertically integrated approaches of the Netherlands and Hungary allowed for greater

integration of new information into policy decisions. In an effort to tackle some of the "wicked problem" characteristics of water resources challenges, researchers have been engaged in developing bottom-up frameworks for decision making with a focus on problems accompanied by deep uncertainty (e.g. Many-Objective Robust Decision Making [Kasprzyk et al. 2013]; Scenario Neutral Planning [Prudhomme, 2010], Decision Scaling [Brown et al. 2012]) with applications developed specifically to aid flood risk decision analysis (e.g. Evers et al. 2018; Knighton et al. 2017a).

Bottom-up decision analysis methodologies initially focus on understanding system vulnerabilities (i.e. What are the negative consequences of a flood that we wish to avoid?), mapping these vulnerabilities onto to a wide range of plausible hazard scenarios (e.g. If an n-year flood occurs, which of the previously defined losses will we experience and possibly to what severity?), and then evaluating which of these hazard scenarios are most likely given our current understanding of atmospheric and hydrological processes. Bottom-up analysis benefits decision making in the ability to avoid contentious discussions about

the reliability of hazard data (e.g. uncertainty in local projections of rainfall intensity under climate change). Such public debates over the "accuracy" of hazard data and risk estimation, of the kind illustrated by recurrent controversies surrounding flood insurance rate maps in the U.S. (Elliott and Rush 2017), reflect a technocratic faith that pegs decision-making to the purported ability of risk analysis to arrive at single true estimates of risk, which models typically do not and cannot provide (Weinkle and Pielke 2017). Vulnerability-based assessments, by contrast, map hazards directly to community vulnerabilities



in order to produce stakeholder-relevant predictions and outcomes, thereby enhancing the broader legitimacy of any subsequent actions taken.

This is not, however, how risk mitigation planning and design has historically been conceptually modeled within the US. Instead, it has been treated as a top-down process, with knowledge transfer between two distinct groups comprised of laypeople

and experts (Wood et al. 2012; Birkholz et al. 2014), the former typically understood as ignorant or "overly emotional" while the latter are presumed to be rational and "analytical" (Lupton 2013). However, this fails to capture how both groups approach issues of risk and natural hazards. As discussed in Norgaard (2011), a "knowledge deficit" model that assumes laypeople would take (rational) action if they "only knew" is too limiting, as it leaves aside the institutional structures (Harries and Penning-Rowsell, 2011) and cultural differences (e.g. Masuda & Garvin, 2006) that shape orientations to risk, institutional

responses, and community vulnerabilities.

Furthermore, the simplistic distinction between professionals and lay people is often blurred in practice. Though the momentum driving flood risk mitigation originates at the federal level within the US (Birby, 2001) and many other nations, implementation of national policies and redistribution of resources relies on social infrastructure at the local level (Few 2003; Rauken et al. 2013; Vogel & Henstra, 2015). Within the US, Canada, Australia, and Europe, flood governance is frequently

the collective effort of organizations operating across scales, including: governmental organizations, non-governmental organizations, privately owned firms, citizen-led groups, and private research organizations (Plummer et al. 2017). Within these organizations, the individuals who participate can be considered experts in one sub-discipline of flood risk mitigation, with shifting leadership roles throughout the process. For instance, governmental organizations may take the lead on policy and legislation, while privately owned firms contribute hydrologic modeling, and residents and citizen-led groups share

knowledge about local vulnerabilities to and effects of exposure to flood risk. These coalitions benefit by leveraging the skills, knowledge, and social position of the varied organizations to more effectively reduce flooding risk.

Bottom-up decision analysis frameworks place a large emphasis on adequately understanding and conveying community vulnerabilities and historical risks into the decision space. These frameworks rely heavily on strong social networks among professionals and the public to bridge gaps among institutions while articulating stakeholder interests (Morss et al. 2005).

Collaboration and stronger trust relationships among institutions and the public have previously led to more effective means of disaster risk mitigation and climate adaptation at the state level (Clarvis & Engle, 2013; Haer et al. 2016). Opportunities to limit cognitive biases should be explored to reduce institutional vulnerability to flooding (Merz et al. 2015), though few research projects have considered how differences among flooding practitioners may be understood or modified to further reduce vulnerability to flooding hazards (e.g. Morss et al. 2005; Downton et al. 2005).

Given the intensity and complexity of the cooperation and coordination required to plan and implement flood risk mitigation, establishing common understandings-of community vulnerability to flooding, baseline flood loss frequency, community willingness, and project desired goals-is a challenging and non-trivial task (Pahl-Wostl, 2009), particularly given diverse backgrounds, education, work experience, and risk exposure of stakeholders found in interdisciplinary working groups. de Brito & Evers (2016) reviewed multi-criteria decision analysis efforts and found that interdisciplinary decision analysis efforts



across multiple stakeholder groups (including both professionals and laypeople) were rare. It is therefore worth re-examining existing social networks and constructs at the local level to determine where such common understandings can be enhanced in the context of bottom-up methodologies for flood risk analysis. This need is particularly relevant when flood risk mitigation planning incorporates climate adaptation goals surrounded by deep uncertainties (Downton et al. 2005; Merz et al. 2015).

We address this gap with a case study survey of 50 professionals working on flood risk mitigation within Tompkins County, New York USA. We define "professional" and "practitioner" here as a subset of flooding risk stakeholders within Tompkins County who have more agency around flood hazard mitigation than that of a community member stakeholder. Our operational definition of professional includes: professionals in private practice and research, elected government officials, appointed government officials, governmental employees, and volunteer members of advocacy groups with a water resources focus. This

focus allows us to understand how the social connectedness among flooding practitioners and their community influences the flood risk mitigation planning and design process. We specifically focus this research on understanding how well positioned this network of flooding professionals is to begin a bottom-up vulnerability-based flood hazard mitigation plan. The results of this research show that "professionals" are not a monolithic category, as they vary in their knowledge of historical hydrologic events, perceptions of existing flooding vulnerability and risk, and perceptions of the need to incorporate future climate

estimates into flood risk mitigation planning and design at the outset of a flood mitigation planning process. We compare survey results to available hydrologic data to determine how classic approaches focused on hydrologic data can be supplemented with socio-hydrologic information, and identify opportunities for strengthening interdisciplinary networks.

## 2 Methodology

### 2.1 Study Region: Tompkins County

Tompkins County lies within central New York, USA. Areas of high population density are clustered within 15 towns and villages, each developed adjacent to a 4th order or higher stream. The county population is approximately 100,000 people across 1300 km$^2$ with a median household income of approximately \$48,000 (US Census, 2017) (about \$10,000 below the 2016 median household income for the U.S.). Tompkins County legislature is presently composed of 10 registered Democrats, 4 registered Republicans, and 1 Independent (TC, 2018), suggesting a Democratic partisan lean.

Federal Emergency Management Adjacency (FEMA) flood insurance rate maps, last updated in 1996, suggest that 3,749 parcels lie within the 100-year special flood hazard area, of which 1,874 are located within the City of Ithaca. From 1978 through 2012, 229 flood loss claims (6.7 claims / year) were submitted through the National Flood Insurance Program (NFIP) totaling \$1,593,201 (~\$46,900 / year) (TC, 2013). The Tompkins County Hazard Mitigation Plan, established in 2013 to review county flood losses and propose corrective actions, is updated annually.

Tompkins County contains four 4th order or greater streams that are tributaries to Cayuga Lake. An active National Weather Service (NWS) flood stage has been established for USGS gage 04234000 (USGS, 2018), which is representative of flooding within the City of Ithaca, a low-lying densely populated community within Tompkins County. Based on annual peak flow



records, exceedance of the 2-m levees or discharge in excess of 120 m$^3$s$^{-1}$ within Fall Creek (adjacent to the City of Ithaca) is estimated to be a 9-year event (Knighton et al. 2017a), though we note that this estimate may vary depending on the period of record the hydrologic data considered.

Tompkins County receives an average of 1,000 mm of precipitation annually, with 15% as snow fall (NCDC, 2018). The
county is approximately 45% forested, 45% agricultural land use (row crops), and 10% urban (Fry et al. 2011). A shallow confining layer (0.5 to 1.5 m) leads to a prevalence of saturation-excess runoff (Easton et al. 2007). Regionally, surface runoff is primarily generated during the spring following extratropical rain-on-snow coincident with frozen or saturated soils and during the fall period of tropical moisture derived precipitation (Knighton et al. 2016, 2017a, b).

Recent trends in gaged streamflow across the Northeast US suggest a more mild increase in extreme discharge relative to the
Conterminous US (Slater & Villarini 2016). Downscaled CMIP5 projections of future precipitation (projected years 2015 – 2100) suggest a slight increase in air temperatures and an associated increase in annual maxima precipitation intensity (Schoof & Robeson, 2016; Ning et al. 2015). Inter-seasonal predictions of future precipitation for Tompkins County show high variability, and potentially inaccurate estimates of seasonal extreme rainfall related to numerical and physical limitations of current General Circulation Models (GCMs) (Wobus et al. 2017; Knighton et al. 2017a). The difficulty in predicting future
extreme precipitation and strong influence of the land surface on flood runoff (Ivancic & Shaw, 2015; Knighton et al. 2017b) has yielded projections of mild to no increase in future riverine flooding hazard in the Northeast US often accompanied by high uncertainty (e.g. Hirabayashi et al. 2013; Wing et al. 2018), or an average increase in risk, but with high spatial variability (e.g. Wobus et al. 2017). Broadly, this region exists with relatively high uncertainty with respect to future climate trends and riverine flood frequency. We anticipate that this lack of a clear signal from state of the art climate and flood projections on the
anticipated future flooding risk may create added difficulty and ambiguity for local decisions concerning the need for local climate adaptation.

### 2.2 Questionnaire Design

Informal interviews were conducted by the authors with ten flooding professionals within Tompkins County from January 2017 through August 2017 to understand what beliefs were commonly held by flood risk mitigation practitioners and which
issues were of most concern. Common themes included: a professional's understanding of where flooding has occurred frequently within the county, a professional's understanding of what socio-economic losses constituted a flood, a professional's perceived community concern about shifting flooding risk under climate change, and potential disagreement among professionals around the design goals of a county-wide flood mitigation project.

We distributed a questionnaire to community members who engage directly with flooding through development of policy and
legislation, science and engineering, education, community outreach, and advocacy. Candidate participants were identified by the Tompkins County Environmental Management Council (EMC), the citizen advisory board to Tompkins County. The EMC's varied experience, long-standing community connections, and formal liaison role between the public, Tompkins County Planning Department, and Tompkins County Legislature allowed it to make informed selections for this study. A



review of the final survey was performed by the Cornell University Institutional Review Board and found to have no ethical implications related to human subjects' participation. A draft of the survey is in the supplementary material.

Survey questions were a mixture of Likert-scale questions, multiple choice selection, and open-ended response. The questionnaire was developed by the authors and piloted with five members of the Tompkins County EMC. The questionnaire

was administered by email on October 27th, 2017 via an online platform. Four survey responses were delivered on paper due to limited access to the internet.

The first goal of the questionnaire was to understand if historical socio-hydrologic data are distributed broadly among flood professionals within Tompkins County. The survey prompted recipients to enter anecdotal information on historical flooding events, specifically: location, date of event, magnitude (i.e. known high water elevations), and known economic losses. The

survey then focused on collecting information on participants' perceptions of current regional flood hazard, risk, and community needs. Individuals were asked to determine what forms of social or economic loss constituted a flood. Individuals were then asked to report their perception of current flooding frequency within their community, and what frequency of flooding would be deemed acceptable. As recommended by Merz et al. (2014) flood studies may benefit from a consideration of how unique weather types and patterns impact flooding. In drawing this explicit link between global climate and local

weather we may better understand potential non-stationary nature of flooding, and how and when climate adaptation should be considered within flood hazard mitigation. The final section of the survey aimed to determine an individual's understanding of how local and regional weather drive flooding within Tompkins County. We asked several questions aimed at understanding flood practitioners' perceptions of community knowledge and desires for climate adaptation planning.

As will be discussed in the results, our survey population was comprised of a relatively small working group of professionals

within Tompkins County, New York (n=50). Our focus on this specific population within Tompkins County led to an inherently small sample size, though the surveyed population was representative of a large proportion of the total population identified (n=89). We therefore used qualitative interpretation of our survey results in place of formal statistical tests.

### 2.3 Hydrometeorological Data Analysis

Weather types and precipitation depth totals for historical events were determined using the daily historical precipitation record

(NCDC, 2018), records of regional historical flooding (Johnson, 1936; Agel et al. 2015; NCDC 2018), a catalog of tropical storms (Roth & Weather Predictions Center 2012), and the personal account of Michael Thorne (City of Ithaca Superintendent of Public Works [personal communication, 2018]) to identify two recent ice-jam events. Return periods for extreme daily precipitation totals were estimated with NOAA Atlas 14 (Percia et al. 2015).

We compare the spatial distribution of flooding as estimated from the FEMA National Flood Hazard Layer (NFHL) 100-year

floodplain (FEMA 2018) and flood practitioner reports.

Historical streamflow records were collected for Fall Creek (USGS 2018) for the period of 1925 to 2018. We use an annual block-maxima approach to identify the significant floods within the publically available long term hydrologic record. We





compare reports of historical flooding from community members to understand how we can best develop a complete record of county flooding.

## 3 Results

### 3.1 Response Rate

The survey was distributed to 89 professionals, of which 50 responded (response rate of 56%). Individuals were asked to self-sort into one of six possible roles: community planning (n=8), education and outreach (n=8), local government leadership (n=9), policy (n=5), advocacy (n=9), and Engineering, Science and Research (ESR, n = 11). We first asked flooding professionals whether they believed they had a good understanding of flood risk mitigation, to which 52% indicated they had a strong grasp of the subject, 40% knew of a professional who could inform them, and 8% were not knowledgeable on the

subject (2 policy, 1 education and outreach, 1 government, and 1 advocacy).

### 3.2 Spatial Distribution of Socio-Economic Flood Losses

Anecdotal reports of flooding were compiled to provide a spatial depiction of commonly flooded locations within Tompkins County (Figure 1), as recalled by research participants. Anecdotal flood reports by community members demonstrate that flooding is a county-wide issue with the most commonly recollected flooding centered on the most densely populated areas.

The reported locations of flooding cover substantially more locations than those falling within the official 100-year special flood hazard area, as depicted on FEMA's flood insurance rate map (FEMA 2018). This is typical of many flood-prone areas; over 20 percent of flood insurance claims come from losses outside of currently mapped high-risk zones (where flood insurance is available but not mandatory) (FEMA, 2015).

NWS flood stage on Fall Creek in Ithaca is estimated to be exceeded with a 9-year recurrence interval. However, reported

dates of flooding events (Table 1) suggest that for much of Tompkins County, professionals have collected information on negative socio-economic consequences from events that are hydrologically more frequent than the 9-year Fall Creek baseline, suggesting that primary sources of hydrologic data alone to not provide a complete depiction of flooding hazards within the county.

Weather types assigned to each reported historical flooding event indicate that flooding has been induced by local extreme

convective precipitation, tropical moisture derived precipitation, extratropical rain-on-snow / snowmelt, and ice-jams. Weather types for events prior to 1930 were not identified due to inconsistency among available sources.

### 3.3 Defining Flooding by Socio-Economic Losses

Exploratory interviews with community leaders suggested that there were 13 socioeconomic losses that individuals commonly used to define a past flooding event (Table 2). The survey presented these 13 possible flooding losses and asked flooding





practitioners to define which types of loss constituted a flood. Professionals also had the option to write in their own preferred definition.

No single type of reported flood was held common to all individuals surveyed (Table 2). The belief that negative flood consequences related to minor erosion in the stream channel and flow above baseflow constituted a flood was only held by a

few respondents. Individuals in planning, government, and advocacy were more likely to hold a broad definition of flooding, whereas individuals in outreach, policy, and ESR tended to hold narrower definition of flooding (Figure 2). About 40% of ESR responses opted to use a write-in definition based on numeric description of flood frequency. For example "any flow exceeding a 100yr or greater storm recurrence interval."

**3.4 Perceptions of Current and Desired Flood Frequency**

Estimates of the current flooding return period for Tompkins County varied slightly by discipline, however, most estimates were below the baseline flood return period established for Fall Creek of the 9-yr event (Figure 3a). The desired reduction in flooding return period varied considerably by discipline. The median ESR, community planning, and outreach response suggests that the expected flood frequency after mitigation efforts should be slightly higher than current flooding hazard

(Figure 3b). The median responses from governmental employees working on legislation and policy desired flood frequency to be reduced to the 100-yr event, suggesting a high level of disagreement between disciplines on anticipated outcomes of flood hazard mitigation. This difference could potentially be due to governmental focus on well-established floodplain thresholds (FEMA 2018) versus perceptions of the physical limits of hydrologic alteration.

There was strong consistency in the perception of current flooding risks (Figure 3a), though the spatial distribution of affected

locations was highly individual (Figure 1). This result suggests that individuals within Tompkins County have a consistent understanding of the frequency of these socio-economic losses; however, there may not be a strong social network for communication of risks as knowledge was spatially constrained by discipline. The Tompkins County hazard mitigation plan contains county records of historical events. This record is presently derived exclusively from federal sources (TC, 2013; NCDC, 2018), with no formal mechanism to collect and archive anecdotal accounts of flooding within the community. In the

absence of a centralized county database to collect and share personal experiences among professionals, an individual's primary spatial knowledge of flooding may be most derived from their own individual experiences, related to place of residence or locations of previous work. Local flood hazard mitigation plans in the US typically suffer from low quality as they are primarily developed as a requirement to maintain access to federal funding instead of functional plans for risk mitigation (Lyles et al. 2014).

We compare aggregated reports of flooding (Figure 1, Table 1) to the long-term historical record of Fall Creek (Figure 4a and b). Results indicate that aggregated records from all community members successfully identify the substantial flooding events where Fall Creek overtopped the 2m levee or greatly exceeded the channel capacity of 120 $m^3s^{-1}$. It is worth noting that prior to 1970 annual peak flow frequently exceeded 120 $m^3s^{-1}$, yet did not exceed the current NWS flood stage of 2 m. This could





be due to a change stream discharge monitoring, or a physical change in the stream rating curve. It was beyond the scope of this study to investigate the cause of the shift in hydrologic response.

The continuous hydrologic data of Fall Creek discharge and stage do not successfully identify all reported flooding events. Though Fall Creek is the largest watershed within Tompkins County (contributing drainage area of 325 km$^2$) and has been

monitored continuously for over a century, a purely hydrologic hazard based assessment considering only primary hydrologic measurements does not provide a complete picture of flooding across the county. The large contributing drainage area (time of concentration ~ 6 hours) results in a hydrologic system sensitive only to weather events on the order of 6 hours or longer. Continuous hydrologic monitoring records contain accurate information at the location of measurement, which is useful for developing flood frequency curves and hydrologic models.  Survey reports of flooding, while less quantitative, benefit from

broader spatial coverage and often include anecdotal accounts of socio-economic losses (e.g., Marjerison et al. 2016). Together, primary hydrologic measurements (e.g. continuous stream depth) and anecdotal survey reports of flooding losses help to develop a more complete picture of the flooding hazard and risk profile within Tompkins County.

### 3.5 Climate-Flood Linkages

Surveyed individuals were asked to report which type of weather mechanism (1 - extratropical system, 2 - local convective

rainfall, 3 - tropical moisture derived rainfall, 4 - snowmelt events, 5 - ice jam) contributed to flooding within Tompkins County with the option to write in flooding mechanisms. Optional write in mechanisms included "Sever [sic] thunderstorms," "dramatic increase in stream levels for any reason," and "extreme rain." Responses and write-in results suggest that there is a general understanding that rainfall and air temperatures relate to flooding events; however, there was no strong agreement within any group that a given weather mechanism contributed directly to flooding outside of county planners agreeing that

local convective rainfall contributed to flooding (Figure 5). This result suggests that practitioners may have a limited understanding of weather-flood linkages.

It is worth noting that on January 11th, 2018, while the online survey was active, a joint snowmelt/ice-jam event caused nuisance flooding throughout the City of Ithaca (Ithaca Times, 2018). We anticipated this specific event would results in a strong agreement among professionals on the relationship between ice-jams, snowmelt and flooding due to recency bias.

However, this was not reflected in the survey results with only 48% (n=24) and 38% (n=19) of all responses suggesting snowmelt and ice-jam release respectively were important flooding mechanisms within Tompkins County.

Anticipation of the need to incorporate climate adaptation into flood risk planning, as well as anxiety around "community perceptions" and "public opposition to planning for climate change" were common themes that emerged during the 2017 informal interviews. Flooding practitioners were asked which direction they anticipated future flooding risk within Tompkins

County would move. The majority of individuals, 30, believed that flooding risk would increase, and 15 responded that they were not sure (Table 3).

Surveyed professionals were asked if they perceived a community desire to implement climate adaptation practices in flood mitigation planning. The result here was less clear, with 18 responding they were not sure, 7 probably not, 15 probably yes,




and 6 definitely yes (Table 4). The Tompkins County Planning Department acknowledges hazards posed by climate change and the need for climate adaptation; however, the current county plan focuses only on maintaining existing natural and built infrastructure. No large-scale flood mitigation projects incorporating climate adaptation currently in planning and design (TC, 2015). There was some disagreement among disciplines on public preference for climate adaptation with ESR and public

advocacy perceiving less interest, and outreach and government perceiving more interest (Table 4).

We next asked practitioners to report their perceptions of the level of climate science knowledge of residents of Tompkins County. Results were divided with 16 responding that they were not sure, 16 believing that residents had basic knowledge, and 10 believe strong knowledge. Results were not substantially different among the disciplines (Table 5).

### 3.6 Optional Write-In Responses

At the conclusion of the survey professionals were given the option to provide any additional information or thoughts on the topic beyond the survey responses provided. We summarize here the results of these submissions. Though we do not aim to interpret these results, they can offer important insights beyond what was captured in the survey questions.

Five professionals supplied optional comments in which they said that they had little knowledge of community perceptions and expressed difficulty in answering these particular questions, with one professional suggesting that community perception

was perhaps too broad to accurately define by one single answer. Three responses suggested that they had a good understanding of community perceptions through involvement with county government and expressed that there was a willingness among the Tompkins County public to involve climate adaptation practices in flood risk mitigation. Four responses attributed recent flooding events to improper control of existing flood mitigation infrastructure by local, state, and federal government. One response listed the ecological benefits of flooding, and suggested that rather than seek mitigation opportunities to control

floods, we seek to adapt human behaviour.

### 4 Discussion

#### 4.1 Definitions and Extents of Flooding

"How do we define a flood event?" and "What are the community vulnerabilities?" appear to be critical questions where establishing consensus may prove difficult. Formal definitions of flooding used within the ESR community, which focus on

quantitative flood frequency, are more hazard oriented, whereas the definitions preferred by planning, government, advocacy, and outreach utilized the socio-economic losses to define flooding. A focus on hazard may simplify engineering design and planning, however, it can potentially be too limiting to properly address other stakeholder needs. A conceptual disconnect on the definition of flooding points to issues with the core problem statement that coalitions of flooding professionals are self-organized to solve.

While formal guidelines exist at the US national level (Obama 2015), these definitions often conflate risk and hazard. Formal federal definitions in the US commonly focus on hydrologic hazard posed by a static water surface elevation, as in FEMA



flood insurance rate maps (FEMA 2018), neglecting hazard associated with discharge velocity, duration of inundation, and suspended material. In addition, the focus on hydrologic hazard can obscure from view the uneven socio-spatial distribution of exposure and vulnerability, which often aligns with prevailing axes of inequality along lines of race and class. The same flood event may be a nuisance for an affluent community and a complete devastation for a poor one. Differences in reported

flooding locations between the survey and established FEMA flood zone maps (Figure 1) could potentially be explained by the distinction between hazard- and vulnerability-based definitions of flooding. For instance, when floods hit particularly vulnerable communities, their impacts may be dramatized in ways that allow them to retain salience in the minds and memories of stakeholders.

Gober & Wheater (2015) propose a broad reconceptualization of flood risk analysis that accounts for emergent and complex

interactions between water and society including: the role of social memory in magnifying risk perception (Di Baldassare et al. 2015) or actual risks (Collenteur et al. 2015), public perception, policy limitations, windows of opportunity, and an imperfect flow of knowledge through society. Such conceptual models may extend unrealistically beyond the capacity of local flood professionals, though we can consider that expanding the definition of flooding beyond the traditional flood frequency realm could allow practitioners to more easily realize the benefits of bottom-up flood risk analysis frameworks. Flooding

practitioners increasingly face decisions about the appropriate level of abstraction when defining socio-hydrology problem statements (Troy et al. 2015; Blair & Buytaert, 2016). Preemptively limiting the complexity of a problem in the planning stage possibly introduces new vulnerabilities in the form of "surprise" (Merz, et al. 2015). It is possible that "surprise" can be avoided or reduced even without consensus on community vulnerability through encouraging inter-disciplinary discourse (DiBaldassare et al. 2016).

**4.2 Perceptions of Climate-Weather-Flood Linkages and Climate Adaptation Planning**

Researchers are moving towards reframing flooding risk from the perspective of the distinct atmospheric mechanisms that induce floods (Merz et al. 2014) in an effort to simplify the interpretation of how meso-scale global trends (e.g. global climate change, decadal global oscillations) influence local weather patterns and subsequently local flooding. Within the Northeast US, this problem is often expressed as non-stationary rainfall intensity and warming air temperatures that may drive flooding

hazards (e.g. DeGaetano, 2009).

Current climate projections of summer extreme precipitation for Tompkins County predict increases in air temperatures and precipitation intensity (DeGaetano 2009) though regional estimates of future flooding hazard are accompanied by high uncertainty (Knighton et al. 2017a). While warming air temperatures are likely to enhance the melt rate of the standing snowpack, it could also limit total snowpack accumulation, tempering the effect of climate change on winter flooding

(Knighton et al. 2017b). Similarly, during the summer season warming air temperatures will likely result in reduced soil moisture, tempering extreme runoff. Though an increase in the intensity of precipitation is often expected to translate directly into increased discharge (e.g. Trenberth, 2011), this outcome is not necessarily expected in the Northeast US (Ivancic & Shaw, 2015).



Emerging methods of flood analysis attempt to limit the need for coarse interpretations of changes to extreme rainfall as projected by highly uncertain GCMs by drawing explicit connections from climate change to weather types and then local flood risk (e.g. Knighton et al. 2017a). While these approaches can potentially better allow local professionals to address climate adaptation within this high uncertainty decision space, they rely on a baseline of knowledge around established climate-

flood linkages. As demonstrated in Table 1, individuals reported historically significant flooding events. Through a simplistic weather-typing analysis, we determined that these events encompassed the atmospheric mechanisms of: extratropical rain-on-snow events, snowmelt, local convective precipitation, tropical storms, and release of ice jams. The reported knowledge of weather systems that induce flooding suggested that professionals did not have a strong grasp on climate-flood linkages, particularly among those engaged in policy development (Figure 5). Many write-in answers indicated that individuals often

did not consider the atmosphere-land surface complexity beyond that of a simple input output system.

Individuals were asked about their beliefs on future flooding risk and two questions pertaining to perceptions of community knowledge and preferences around climate adaptation planning. First, responses indicated that approximately 1/3 of all professionals surveyed were unsure about community perceptions of climate science and the importance of implementing climate adaptation practices into flood hazard mitigation. Practitioners agreed that communities had some understanding of

climate science that was either basic or good. Practitioners disagreed on the community desire to incorporate climate adaptation practices into flood mitigation planning, which may reflect the politically contentious nature of climate change in U.S. political discourse and policymaking more generally. Previous research has demonstrated disconnections between public climate-flood risk perceptions and expert opinion (e.g. Hamilton et al. 2016; Ogunbode et al. 2018; Shepard et al. 2018). In this case study, professionals disagreed on both community perceptions (Table 4) and climate education (Table 5). While there may be

conceptual differences in the beliefs of laypeople and experts, we possibly take for granted that experts effectively understand and represent complex community needs and beliefs in the flood mitigation planning process.

**4.3 Perceptions of Flooding Expertise and Social Networks of Professionals**

As previously described, 52% of flooding professionals within Tompkins County reported that they had a good understanding of flood risk mitigation, and 40% reported that they did not, but had a resource who could inform them, suggesting a well-

connected network among professionals. Survey results indicate strong disagreement among flooding professionals as to which socio-economic losses constitute a flood (Table 2), incomplete knowledge of the spatial extent of flooding within Tompkins County (Figure 1), disagreement on anticipated community needs (Figure 3b), weak understanding of climate-weather-flood linkages (Figure 5), and some disagreement on community perceptions on climate adaptation (Tables 4 and 5).

The results of this research suggest a case where practitioners may believe that they are well-informed and share commonly

held beliefs, while in reality the network of flooding professionals is less well established or hold divergent perceptions or terminologies regarding flooding. Instances where decision-makers believe their perspective is commonly held can open problematic possibilities and new vulnerabilities. Flood risk is a particularly difficult problem to address owing to the infrequent nature of hydrologic extremes, non-linear relationship between rainfall, runoff, and exposure, and potential socio-





economic feedbacks that develop between society and flood hazards (e.g. the "levee effect" [Collenteur et al. 2015]). The large number of professionals who reported that they have a strong grasp of the subject could potentially indicate a susceptibility to cognitive biases influencing flood mitigation planning and design (Merz et al. 2015). For example, practitioners may be particularly susceptible to over-confidence and confirmation bias with respect to their currently held understanding of existing

flood risks.

In developing the current estimate of flood frequency from aggregated reports of historical events we reached a relatively accurate appraisal of flooding hazard (Figures 1 and 4). This result demonstrates that flood mitigation can benefit strongly if the knowledge of independent institutions is properly leveraged (e.g. Llobet et al. 2016). In other cases, this research identifies gaps in the social networks of flooding professionals. Inflexibility among professionals to consider variations in project goals

or risk tolerance can lead to undesirable flood mitigation outcomes (Downton et al. 2005). Disagreement on the definition of flooding and community preferences for flood mitigation are perhaps expected results at the inception of a flood mitigation project and present opportunities for flooding professionals to engage more directly across disciplines.

Implementation of emerging robust decision making frameworks (e.g. Prudhomme 2010; Brown et al. 2012) commonly dictate that a bottom-up approach be taken with regard to defining the problem and weighting desirable outcomes. These approaches

place significant emphasis on understanding community vulnerability (Pielke et al. 2012) as the initial objective. Our results suggest that professionals engaging in flood mitigation within Tompkins County are in agreement about exposure (Figure 3a), but have some disagreement around flood vulnerabilities (Figure 1) and anticipated needs (Figure 3b). Substantial differences emerged in how the six groups of professionals responded to certain questions. ESR preferred to define flooding from a classic perspective focusing on hydrologically relevant metrics (e.g. the n-year discharge and exceedance of bank-full discharge),

whereas individuals in planning, government, and advocacy defined flooding very broadly from a bottom-up perspective of the associated socio-economic losses. This particular result is not surprising as there is substantial variation in professional norms among the participating flood mitigation roles owing to education, experience, and their respective audiences/constituencies.

Distinct institutions working on flood mitigation may engage differently with primary and secondary data sources. For

example, engineering, science and bio-physical research groups are typically focused on the collection and interpretation of meteorological and hydrologic data, while those engaged in public outreach may be more responsible for the collection and interpretation of socio-hydrologic data (i.e. historical flooding economic losses, migration), and governmental organizations may best understand policy implications of flood investments. The organization of social networks of professionals working in the realm of flood risk reduction remains a fairly unstudied subject, though particularly relevant as communication of highly

uncertain information among professional disciplines remains a challenging task (e.g. Pappenberger et al. 2013; Morss et al. 2005; Downton et al. 2005). Social organizations may lead to the development of risk perception networks, where clusters of individuals share the same risk perception (Scherer & Cho, 2003). Conversely, dissimilar groups may tend to develop less connected relationships, where information quality is only high within a subset of organizations and shared only when



necessary. Finally, disagreement surrounding the role of primary data and expression of hydrologic uncertainty can lead to suboptimal solutions or inaction (e.g. Downton et al. 2005).

Independently operating institutions at the local level commonly hold individual goals (Butler & Pidgeon 2011; Measham et al. 2011), which may be distinct from the collective flood risk mitigation goal. Given their orientation and obligations toward

voters and taxpayers, governing organizations are commonly driven by a desire for continued local economic (re)development (Molotch 1999) as well as preoccupied with concerns around "blame" and "credit" for social outcomes (Leong & Howlett 2017). This sometimes results in a narrow temporal focus and decisions that are made with limited consideration for decadal or longer process (Gober & Wheater 2015). Private firms may be concerned with maintaining profitable contracts, job security, ethics, and liability. NGO and advocacy groups may be concerned with developing and maintaining public interest (Lorenzoni

& Pidgeon 2006). Research organizations are often concerned with developing new science and engineering techniques, with a tendency to avoid advocacy and maintenance of impartial stances on controversial subjects (Singh et al. 2014).

Disconnections among the network of professionals could also be related to compartmentalization of urban and non-urban problems. Tompkins County contains one urban center (City of Ithaca) and 14 less densely populated towns and villages. Observed differences in flooding perception among professionals could stem from the differences in community structures

and land use within Tompkins County. Urban fill substantially alters the hydrologic response of urban areas, potentially confounding the relationship between rainfall and runoff (Knighton et al. 2014) that is not experienced in less developed areas with native soils. We note that reports of flooding and flood loss claims are greatest within City of Ithaca (TC, 2013) possibly suggesting that a combination of regional hydrology and population density is driving the perception of flooding risk and the need for mitigation within the City of Ithaca; however, Marjerison et al. (2016) through a broader spatial study suggest that

local population density may not be a sufficient regional factor to determine flooding frequency perception. Beyond density, urban vulnerabilities and problems are often considered to be distinct from non-urban areas due to greater socio-political institutional complexity and less deliberate community planning (Zevenbergen et al. 2008).

Strong networks of professionals have been demonstrated to simplify decision making even in challenging situations (e.g. Bracken et al. 2016). Efforts to refine the flood hazard and risk profile of Tompkins County that make explicit the effort to

involve members of all disciplines may build trust and communication among practitioners (Morss et al. 2005) and the public. Work that strives to leverage the knowledge of independently operating institutions and the public to define the problem statement and guide mitigation practices ultimately improves (e.g. Serra-Llobet et al. 2016; Edelenbos et al. 2017). Public willingness to take risk reduction measures has previously been attributed myriad variables including actual risk and societal norms (Lo 2013), income (Lo 2014), exposure to or protection from past events (Di Baldassare et al. 2013), and trust in expert

opinion (Wachinger et al. 2013). Though disagreement between flooding practitioners and the public may occur, establishing these social connections could be an important step towards establishing trust, building public support for mitigation projects, and opening opportunities to collect socio-hydrologic data which could improve flood mitigation planning and design.



### 4.4 Broader Impacts

European nations (e.g. Næss, et al. 2005), the UK (e.g. Brown & Damery 2002), African and Asian nations (e.g. Huntjens et al. 2011), and the US face the prospect of enacting water-governance and developing policy within a changing climate. Recent research has focused heavily on the shortcomings of top-down approaches to flood hazard mitigation as enacted by a variety

of governmental organizations. For example, Brown & Damery (2002) explore the structural issues present in a top-down governance scheme within the UK and conclude that a focus on hazard lead to improper problem definitions and "undersocialized" solutions. Subsequent research has proposed that governance leveraging both top-down and bottom-up schemes could improve the efficiency with which a nation incorporates societal vulnerability information into policy (e.g. Pahl-Worstl et al. 2010), thereby lowering societal risk.

Recent applications of decision analysis frameworks for flood mitigation within Europe (e.g. Evers et al. 2018) and the US (e.g. Knighton et al. 2017a) highlight the technical potential of these approaches, yet as a research community, we have not fully explored these frameworks outside of a handful of carefully controlled case studies. As more attention is being given to bottom-up approaches as a potential panacea for flood hazard mitigation, a critical assessment of governmental organizations, institutions, and practitioners becomes more necessary to explore possibilities for new unforeseen vulnerabilities that may

emerge.

While our research focuses exclusively on a network of professionals within Tompkins County New York (US), aspects of our work can provide broader insights. First, the local governmental and institutional organization of this case study mirrors that of other US and European cities, which suggests the possibility of similar institutional vulnerabilities associated with local governmental-, private-, and community-organizations. For example, Bracken et al. (2016), studying flood management within

the UK, describe a similarly loose coalition of experts from governmental and non-governmental organizations to those observed within our research.

Second, the reliance of bottom-up decision analysis frameworks on networks of people exists independent of local governmental structure, and would likely contribute similar vulnerabilities as we have observed. Merz et al. (2015) review a series of historical floods across Europe that resulted in increased devastation as a result of "surprise." Surprise is then

attributed by Merz et al. (2015) to cognitive biases "hardwired in the human brain." It is possible that the divergent perceptions and definitions that we observe among Tompkins County professionals are indicative of universal human traits rather than simply a local phenomenon.

Finally, our methodology is easily adapted, and could be applied to uncover new vulnerabilities in parallel governmental structures in other nations.

### 30 5 Conclusion

Complex decisions involving highly uncertain inputs can potentially be reduced to more manageable problems given the recent improvements to and wider application of bottom-up vulnerability-based decision analysis frameworks. This family of





methodologies relies heavily on precise definition of system vulnerabilities so that uncertain inputs (e.g. climate projections of extreme rainfall) can be readily mapped to a stakeholder-relevant metric of concern. In flood risk analysis these frameworks are proving particularly useful as they may help to avoid debates around the reliability of future climate projections by placing this information in the context of sensitivity to stakeholder-relevant outcomes. In cases where risk sensitivity to climate

variability is low, discussions on the reliability of climate projections can be readily avoided. Further, there is an emerging consensus that flood risk analysis should directly consider the unique characteristics of the atmospheric mechanisms that induce flooding. Research advances have demonstrated that explicitly separating storms by weather types allows for stronger inferences on future flooding hazards.

Flood hazard and risk mitigation in the US is often carried out at the local level by an informal collection of governmental,

non-governmental, private, and academic institutions. Previous research has observed variations in how institutions approach flood hazard owing to differences in educational and professional backgrounds, variations in their relationship to socio-hydrologic data, and which goals or outcomes are deemed desirable. Given these discrepancies, there is a strong need to review differences among disciplines to ensure that emerging bottom-up vulnerability-based frameworks can be readily incorporated into local planning efforts.

Our research demonstrates that there are broad differences in belief among practicing professionals within Tompkins County, NY as to what socio-economic losses constitute a flooding event and spatially disaggregated knowledge of historically flooded locations. There was strong agreement on the frequency of flooding experienced by residents, but disagreement around the desired level of protection from a flood mitigation effort. These results suggest that there is some variance among flooding professionals on the definition of community vulnerability to flooding. Undisclosed or unknown disparities in perceptions

among flooding practitioners could serve as barriers to successfully implementing vulnerability-based frameworks for decision analysis. Developing strong definitions of flooding vulnerability may not require explicit agreement among all practitioners, but rather venues that allow for productive processes of deliberation. These venues necessarily involve multiple, diverse stakeholders whose input shapes an outcome that all parties can agree is procedurally fair and acceptable.

Explicit consideration of climate-flood linkages showed similar barriers based on practitioners' knowledge. Professionals

identified historical events induced by five unique mechanisms, but failed to identify these types of weather events as important causes of local floods. Among the survey results there is an intuitive sense that intense rain causes flooding, though it is possible this limited understanding prevents conceptual connections of local events to regional and global climate patterns. These discrepancies could serve as a barrier to implementing important advances in flood risk engineering that aim to use relevant climate projections to inform local planning.

In aggregate, the knowledge base of the Tompkins County flood practitioners served to provide a well-defined picture of community vulnerability and perceptions, though the beliefs of individuals varied. Previous research suggests that collaborative efforts can work to improve connections between social networks of experts and laypeople. This research demonstrates the need for interdisciplinary research, planning, and design throughout flood risk mitigation and climate adaptation planning to maintain strong social connections not just between laypeople and experts, but among experts.



**6 Acknowledgements**

This research was supported by an Engaged Opportunity Grant from the Cornell University Office of Engagement Initiatives.
We acknowledge the contributions of the Tompkins County Environmental Management Council in identifying flood hazard
mitigation practitioners within Tompkins County. We specifically thank Michael Thorne (City of Ithaca Superintendent of
Public Works) and Scott Doyle (Tompkins County Planning Department) for their guidance on this research.

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



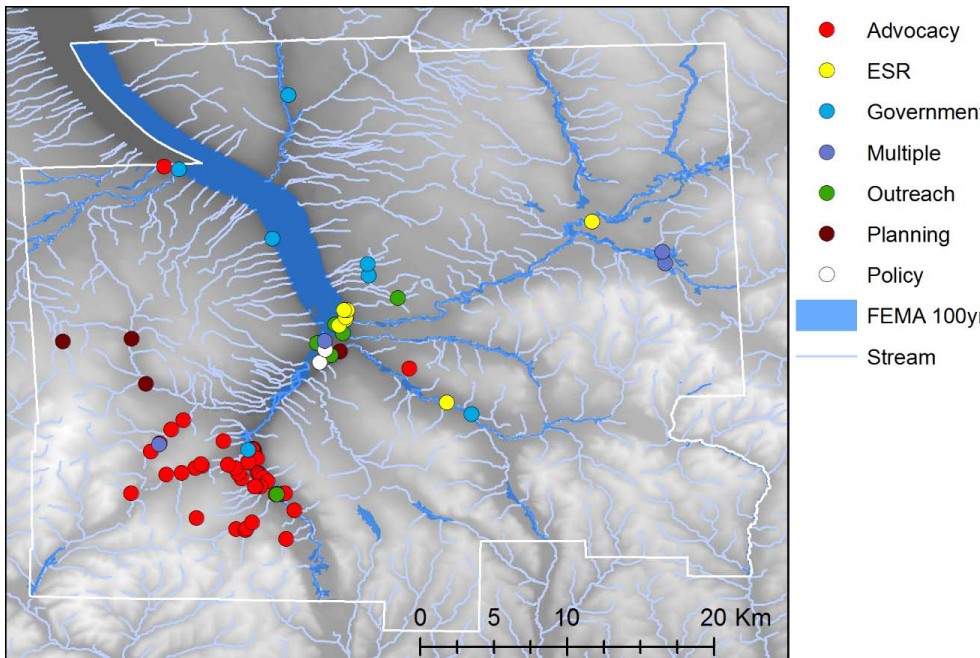

**Figure 1: Spatial distribution of survey-reported flooding within Tompkins County (filled circles) and FEMA 100-yr flood plain (dark blue)**





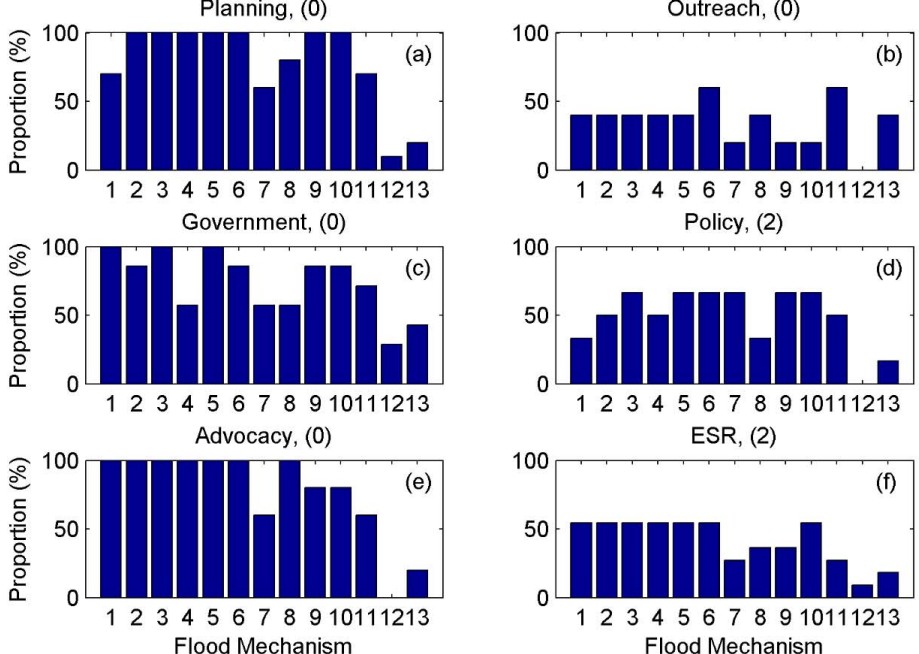

**Figure 2: Socio-economic losses that defined flooding events by discipline (Table 2 subset by discipline). Values in parenthesis indicate the number of respondents who did not offer an answer.**



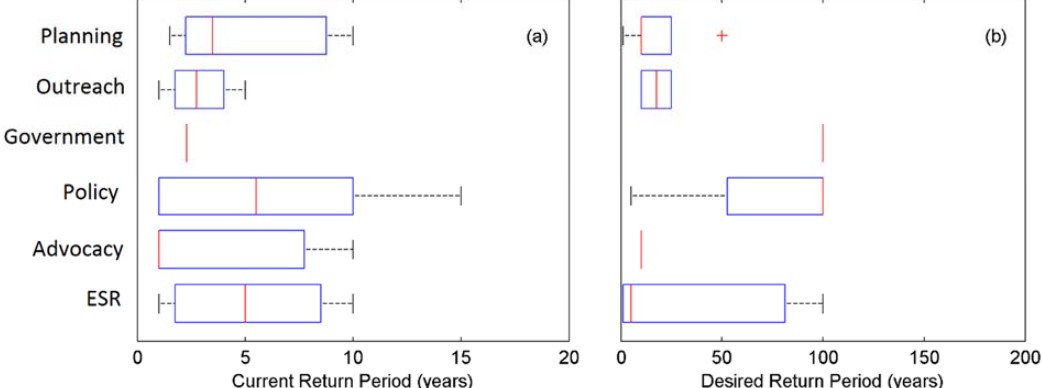

**Figure 3: Estimated a) current flood loss return period and b) desired return period resulting from flood hazard mitigation efforts. Reports of Desired Frequency above 100 years are presented as 100-yr for visualization. Red lines indicate the median.**



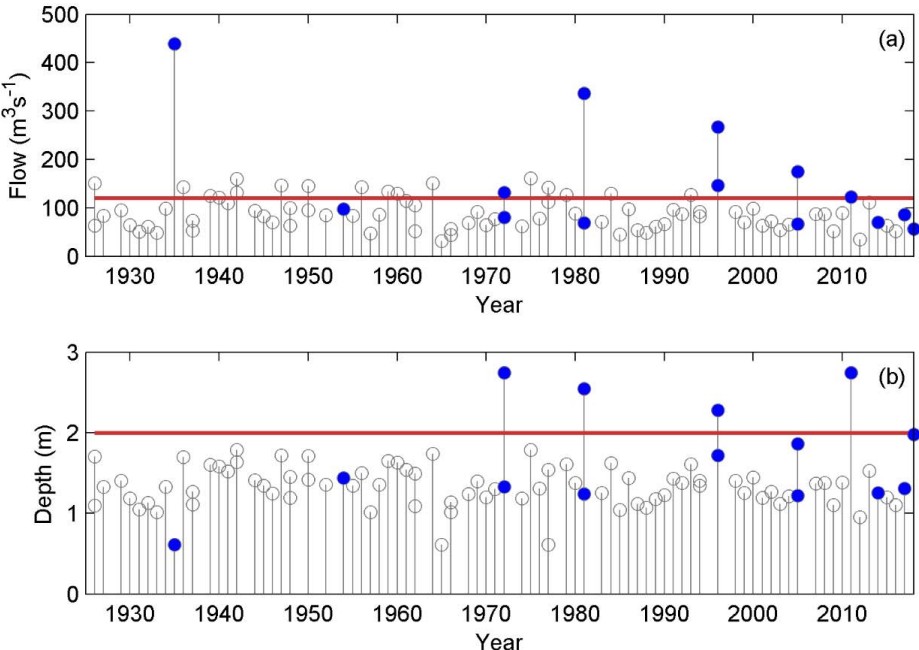

**Figure 4: Annual block maxima a) peak instantaneous discharge and b) peak stage for Fall Creek. Blue dots indicate that the event was identified as a flood by at least one survey response. Red lines indicate the hazard thresholds for a) exceeding channel capacity, and b) overtopping the 2 m levee within the City of Ithaca**





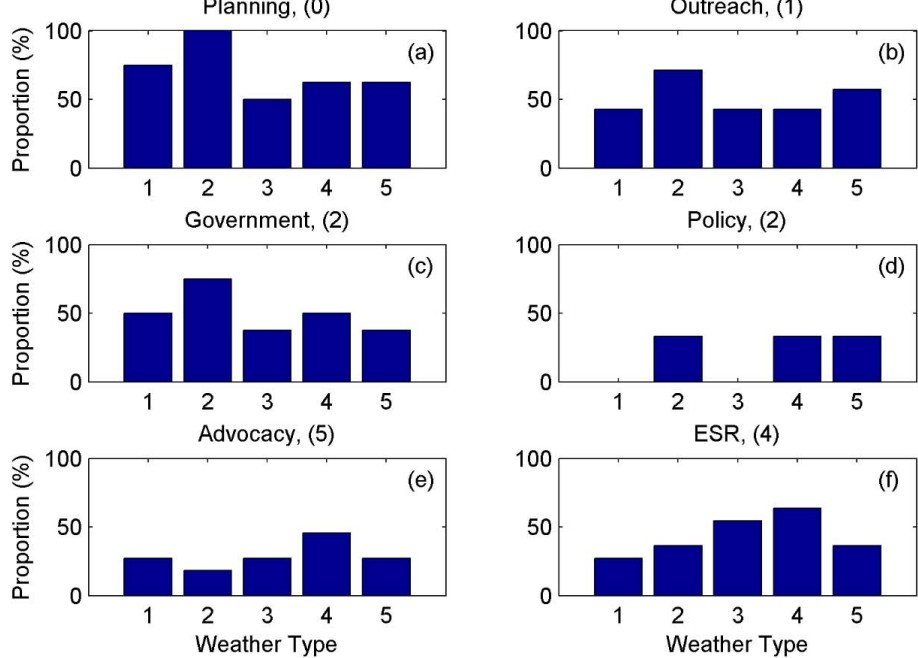

**Figure 5: Reported flooding mechanisms that contribute to local flooding. 1 - extratropical system, 2 - local convective rainfall, 3 - tropical moisture derived rainfall, 4 - snowmelt events, 5 - release of ice jam. Values in parenthesis indicate the number of respondents who did not offer an answer.**





**Table 1: Reported historical flooding events. Rainfall totals are the maximum daily precipitation (NCDC, 2018). Return periods are determined from NOAA Atlas 14 (Percia et al. 2015).**

[a] Statewide flooding was reported to result from a mixture of a Tropical Moisture Export and local convective rainfall

| Date | Rainfall (cm/day) | Return Period (yrs) | Weather Type |
|---|---|---|---|
| 4/18/1905 | 1.2 | < 1 | - |
| 6/3/1905 | 4.3 | < 1 | - |
| 6/17/1905 | 4.6 | < 1 | - |
| 7/3/1905 | 4.8 | < 1 | - |
| 7/8/1935 | 20.0 | > 1000 | Tropical/Local convective rain[a] |
| 11/3/1954 | 4.0 | < 1 | Hurricane Hazel |
| 6/23/1972 | 9.0 | 10 | Hurricane Agnes |
| 10/28/1981 | 12.9 | 25 | Local convective rain |
| 1/19/1996 | 4.7 | < 1 | Rain on snow |
| 4/3/2005 | 5.7 | 2 | Rain on snow |
| 9/8/2011 | 11.3 | 25 | Tropical Storm Lee |
| 1/11/2014 | 0.0 | < 1 | Ice jam |
| 6/14/2015 | 10.4 | 10 | Local convective rain |
| 7/1/2017 | 0.9 | < 1 | Local convective rain |
| 1/12/2018 | 2.4 | < 1 | Ice jam |



**Table 2: Results of which socioeconomic losses were considered a flooding event. Types 14 and 15 are write in responses**

| Type | Description of Flood | Number of Responses |
|---|---|---|
| 1 | Loss of life | 29 |
| 2 | Damage to private structures | 32 |
| 3 | Displacement of people | 34 |
| 4 | Damage to vehicles | 30 |
| 5 | Damage to public property | 34 |
| 6 | Inundation of public roads | 34 |
| 7 | Flow over private property | 21 |
| 8 | Backed up culverts | 25 |
| 9 | Loss of streamside vegetation | 29 |
| 10 | Stream flow out of channel banks | 32 |
| 11 | Substantial erosion in the stream channel | 24 |
| 12 | Minor erosion in the stream channel | 4 |
| 13 | Any flow greater than baseflow | 11 |
| 14 | Discharges above an $n$-year recurrence interval | 4 |
| 15 | Any negative impact to resources | 1 |





**Table 3: Perceptions of future riverine flooding risk within Tompkins County by flooding practitioners**

|  | Not Sure | Less Risk | Same Risk | More Risk |
|---|---|---|---|---|
| Community planning | 0 | 0 | 0 | 8 |
| Education and outreach | 1 | 0 | 1 | 5 |
| Local government leadership | 2 | 0 | 1 | 5 |
| Policy development | 3 | 0 | 0 | 2 |
| Public advocacy | 3 | 0 | 0 | 6 |
| ESR | 4 | 0 | 3 | 4 |
| Total | 15 | 0 | 5 | 30 |



**Table 4: Perceptions of community desire to implement climate adaptation planning in flood risk mitigation**

|  | Note Sure | Definitely Not | Probably Not | Probably Yes | Definitely Yes |
|---|---|---|---|---|---|
| Community planning | 2 | 0 | 1 | 5 | 2 |
| Education and outreach | 1 | 0 | 0 | 4 | 0 |
| Local government leadership | 1 | 0 | 1 | 4 | 1 |
| Policy development | 5 | 0 | 0 | 0 | 1 |
| Public advocacy | 2 | 0 | 2 | 1 | 0 |
| ESR | 5 | 0 | 3 | 1 | 2 |
| Total | 18 | 0 | 7 | 15 | 6 |



**Table 5: Perceptions of general community knowledge level of climate science and adaptation**

|  | Not Sure | Little Knowledge | Basic Knowledge | Strong Understanding |
|---|---|---|---|---|
| Community planning | 1 | 0 | 3 | 5 |
| Education and outreach | 1 | 0 | 3 | 1 |
| Local government leadership | 1 | 2 | 2 | 2 |
| Policy development | 5 | 0 | 1 | 0 |
| Public advocacy | 1 | 1 | 2 | 1 |
| ESR | 5 | 0 | 5 | 1 |
| Total | 16 | 3 | 16 | 10 |