# Peer review of "Challenges to Implementing Bottom-Up Flood Risk Decision Analysis Frameworks: How Strong are Social Networks of Flooding Professionals?"

_Hydrology and Earth System Sciences, 2018_

## Referee Comment (RC1) · Anonymous Referee #1 · 6 Sep 2018

The paper deals with and interesting and novel aspect in flood risk management research. Due to insights that identification of vulnerability is of utmost relevance for actual flood risk reduction and the fact that vulnerability is very context and spatial depended the paper contributes with a valuable contribution to this research field. Here the vulnerability is defined by stakeholders which reveals different understandings and relevance of criteria of flood vulnerability. The overall conclusions are clear and the presentation is well structured. However, some parts of the text are a bit unclear and the methods description needs some revisions. , I suggest the following considerations

for improving the manuscript: - Please explain what type of floods you are dealing with. It is important to differentiate river floods, flash floods etc. - Page 1, line 29: please explain imperfect understanding. How can understanding be imperfect? - Page 2, line 10: probably more recent literature is available - Page 2, line 29: please explain how bottom-up analysis benefits decision making. Per se or what is necessary for better dm? - Page 5, line 23: what is an informal interview. That is not clear to me. Please explain this methodological approach. - Page 8, line 1: only professionals? - Page 9, line 25: why "only"? 48% is not a little number.

---

## Referee Comment (RC2) · Anonymous Referee #2 · 10 Sep 2018

Based on the case study of the Tompkins County, the paper helps for understanding the social network of flooding professionals is to begin a bottom-up vulnerability-based flood hazard mitigation plan. The work is valuable and interesting. I think the paper is likely worth publishing.

Comments: 1. The study is conducted through a case study in the Tompkins County, the population of which is approximately 100,000 people across 1300 km2. Although the authors have discussed broader impacts of their bottom-up decision analysis frameworks, a more complete and clear discussion about the scale of the framework

could be used. Is it applicable for other larger or smaller counties? How can the framework be "bottom-up" to states?

2. It mentions that the Tompkins County is a community dealing with moderate flooding. So, is the framework applicable for other communities dealing with other types of flooding?

3. Page 7 Line 9: 8% of 50 responded professionals are 4, but (2 policy, 1 education and outreach, 1 government, and 1 advocacy).

---

## Author Comment (AC1) · 14 Sep 2018

We thank reviewer 1 for their comments on this research. Please find our responses to specific comments below.

- The paper deals with and interesting and novel aspect in flood risk management research. Due to insights that identification of vulnerability is of utmost relevance for actual flood risk reduction and the fact that vulnerability is very context and spatial depended the paper contributes with a valuable contribution to this research field. Here

[Figure]

the vulnerability is defined by stakeholders which reveals different understandings and relevance of criteria of flood vulnerability. The overall conclusions are clear and the presentation is well structured. However, some parts of the text are a bit unclear and the methods description needs some revisions. , I suggest the following considerations for improving the manuscript:

- Please explain what type of floods you are dealing with. It is important to differentiate river floods, flash floods etc.

The size of the largest catchment (325 sq. km) is small enough that the stream response typically occurs within 6 hours of the onset of precipitation. Still, the flooding occurs when natural levees are overtopped. I believe this case could be classified as both flash flooding and riverine flooding. Because the risk is primarily related to the flood stage elevation and not a lack of warning time, we will classify this problem as riverine flooding.

We have revised the introduction as follows:

Page 1, Line 26: "Societal vulnerability to riverine flooding is a complex function of physical hydrological processes"

Page 2, Line 3: "Riverine flood risk analysis is inherently difficult due to the infrequency of flooding events"

Page 4, Line 5: "We address this gap with a case study survey of 50 professionals working on riverine flood risk mitigation within Tompkins County, New York USA."

Also, see mention of riverine flooding in revision for item #2.

- Page 1, line 29: please explain imperfect understanding. How can understanding be imperfect?

We have revised this sentence as follows: "…have been exacerbated by uninformed and inaccurate prior beliefs surrounding riverine flood hazards…"

[Figure]

- Page 2, line 10: probably more recent literature is available

We agree. We have also included the following reference which provides detailed definitions of top-down and bottom-up governance.

Serra-Llobet, A., Conrad, E., & Schaefer, K.: Integrated water resource and flood risk management: comparing the US and the EU, E3S Web of Conferences, 7, doi:0.1051/e3sconf/20160720006, 2016.

- Page 2, line 29: please explain how bottom-up analysis benefits decision making. Per se or what is necessary for better dm?

We agree that our point was not clear. We have revised the introduction as follows:

"Flood decision making can be stalled by contentious discussions about the reliability of hazard data (e.g. Is climate change driving changes to local storms? Should climate change be accounted for in mitigation planning?). Bottom up decision making frameworks benefits the process in that uncertain data and potentially controversial methodologies can be evaluated within the context of community risks. For example, climate change driven changes to storm intensity may not increase frequency severe economic losses, and therefore can possibly be disregarded. Such public debates over the "accuracy" of hazard data and risk estimation, of the kind illustrated by recurrent controversies surrounding flood insurance rate maps in the U.S. (Elliott and Rush 2017), reflect a technocratic faith that pegs decision-making to the purported ability of risk analysis to arrive at single true estimates of risk, which models typically do not and cannot provide (Weinkle and Pielke 2017)."

- Page 5, line 23: what is an informal interview. That is not clear to me. Please explain this methodological approach.

We agree. We now provide a reference and a more descriptive title for this technique "semi-structured interviews." We have modified the text as follows:

"We conduct semi-structured interviews (methodology described by Hermanowicz,
2002) with ten flooding professionals within Tompkins County from January 2017 through August 2017. Each interview was initiated with a series of general questions on the topic of flooding, and shortly thereafter interviewees were encouraged to move the discussion in their own direction of interest. The purpose of these interviews was to develop an exhaustive inventory of themes concerning the challenges faced by professionals engaging in group decision making and ideas about flooding commonly held by flood risk mitigation practitioners."

- Page 8, line 1: only professionals?

We have modified this line as follows: "Surveyed individuals also had the option to write in their own preferred definition."

- Page 9, line 25: why "only"? 48% is not a little number.

We disagree slightly with this comment. The word "only" was included because we anticipated a result closer to 100% because of recency bias. A moderate flood happened throughout the county which was widely reported by local media. We anticipated that individuals engaging with flooding work on a professional level would have taken notice of this event. That only 48% identified snowmelt as a critical mechanism was quite surprising, and importantly points to a disconnection between actual hazards, and those designing mitigation practices.

---

## Author Comment (AC2) · 14 Sep 2018

We thank reviewer 2 for their specific comments on our work, particularly the comments related to the framing of our research.

- Based on the case study of the Tompkins County, the paper helps for understanding the social network of flooding professionals is to begin a bottom-up vulnerability-based flood hazard mitigation plan. The work is valuable and interesting. I think the paper is likely worth publishing.

[Figure]

- Comments:

- 1. The study is conducted through a case study in the Tompkins County, the population of which is approximately 100,000 people across 1300 km2. Although the authors have discussed broader impacts of their bottom-up decision analysis frameworks, a more complete and clear discussion about the scale of the framework could be used. Is it applicable for other larger or smaller counties? How can the framework be "bottom-up" to states?

- 2. It mentions that the Tompkins County is a community dealing with moderate flooding. So, is the framework applicable for other communities dealing with other types of flooding?

We are grouping our response to these comments 1 and 2 as they cover similar issues. First, we agree with the reviewer that more elaboration will allow the paper to have a broader impact. We have revised the broader impacts section as follows:

"It is worth discussing that several aspects of our study catchment may have implications for how these results can be interpreted and applied to other locations. Our research focuses exclusively on a network of professionals within Tompkins County New York (US), with a distinctly bottom-up structure for flood governance. The county is moderately sized (population of 100,000) and experiences moderate flooding ($\sim$9 year recurrence interval for socioeconomic riverine flood losses).

With respect to county population and flood loss frequency, we can possibly anticipate that the social connectedness of professionals would increase with increasing community size and flood frequency. Both larger populations and increased frequency of hazards could lead to more complete records of historical floods and increased among professionals. It is possible that increased exposure to flood frequency would reduce cognitive biases (e.g. Merz et al 2015) leading to an "adaptation effect" (Di Baldassarre et al 2015). Conversely, less exposure as would be expected with a smaller population and less flooding risks would be expected to decrease social understanding of flooding

risks (Collenteur et al. 2015) and less established networks of professionals.

Broadly, there are several aspects of this research which may allow our results to be more globally applicable. First, the local governmental and institutional organization of this case study mirrors that of other US and European cities, which suggests the possibility of similar institutional vulnerabilities associated with local governmental-, private-, and community-organizations. For example, Bracken et al. (2016), studying flood management within the UK, describe a similarly loose coalition of experts from governmental and non-governmental organizations to those observed within our research. Second, the reliance of bottom-up decision analysis frameworks on networks of people exists independent of local governmental structure, and would likely contribute similar vulnerabilities as we have observed. Merz et al. (2015) review a series of historical floods across Europe that resulted in increased devastation as a result of "surprise." Surprise is then attributed by Merz et al. (2015) to cognitive biases "hardwired in the human brain." It is possible that the divergent perceptions and definitions that we observe among Tompkins County professionals are indicative of universal human traits rather than simply a local phenomenon. Finally, our methodology is easily adapted, and could be applied to uncover new vulnerabilities in parallel governmental structures in other nations."

- 3. Page 7 Line 9: 8% of 50 responded professionals are 4, but (2 policy, 1 education and outreach, 1 government, and 1 advocacy).

Yes, this was a typo. The correction reads: "10% were not knowledgeable on the subject..."
* * *